# Post-Cryogenic Viability of Peach (*Persica vulgaris* Mill.) Dormant Buds from the VIR Genetic Collection

Vladimir Verzhuk [1,*], Victor Eremin [2], Taisya Gasanova [2], Oksana Eremina [2], Liubov Y. Novikova [1], Galina Filipenko [1], Maxim Sitnikov [1] and Alexander Pavlov [1]

[1] N.I. Vavilov All-Russian Institute of Plant Genetic Resources, 42, 44 Bolshaya Morskaya Street, St. Petersburg 190000, Russia
[2] N.I. Vavilov All-Russian Institute of Plant Genetic Resources, Krymsk Experiment Breeding Station of VIR, 12, Vavilova St., Krymsk 353384, Russia
* Correspondence: vverzhuk@mail.ru

**Abstract:** The long-term storage of the genetic resources of fruit crops for breeding needs can be freely developed by cryopreservation cuttings with dormant buds in liquid nitrogen vapor, but so far, this method has not been practically used for peach. Cuttings with dormant buds of five peach varieties growing in the field gene bank at Krymsk Experiment and Breeding Station of VIR were collected for cryopreservation in 2019–2021. The three-factor analysis of variance showed that the viability of peach cuttings was significantly affected by the year ($p < 0.001$) and variety ($p < 0.001$). According to the three-year average characteristics of the cultivars, the analysis of variance showed a significant difference in the viability of the cultivars after cryopreservation ($p = 0.004$). According to the results of the three years of study, cvs. 'Podarok Kryma' (43.3%) and 'Lucky 24 B' (44.4%) showed the lowest viability after cryopreservation, significantly lower than cvs. 'Baby Gold' (54.4%) and 'Ustojchivy 90' (55.6%). Cv. 'Lyubimets Krasnodara' (48.9%) occupied an intermediate position. These viability values exceeded the minimum requirement for samples subjected to long-term cryogenic storage in a cryobank. Low-temperature storage of peach cuttings at –5 °C can be used for short-term preservation. After low-temperature storage, the viability of peach cutting amounted to an average of 67.1%.

**Keywords:** low-temperature storage; cryopreservation; plant genetic resources; peach; liquid nitrogen vapor

## 1. Introduction

Peach—Prunus persica (L.) Batsch (=Persica vulgaris Mill.)—is a stone-fruit crop widespread in southern Russia due to its plasticity. The area of its commercial cultivation is considered to be a temperate zone between 45° N and 30° S. Peach belongs to the plum genus (Prunus L.); it is a fruit tree with a lush, dense crown with a diameter of about 6 m and a height of 8 m. The plant is supposedly native to northern China [1].

The Krymsk Experiment and Breeding Station of the All-Russian Institute of Plant Genetic Resources (VIR) maintains a collection of stone fruits that is the largest and most important in Russia and any part of the former Soviet Union. Three sites at the station are occupied by plantings of collection samples of peach and nectarine. One was planted in 2004, the other in 2014. At the third site, located at an altitude of 150 m above sea level, the entire collection of peach is currently transplanting, numbering 398 accessions of various ecological and geographical origin of which 109 accessions are introduced species and 289 are domestic varieties. Thirty-five genotypes are varieties and selection samples of the Krymsk Experiment and Breeding Station. With the instability of climatic, economic, and ecological conditions, there is always a threat of losing valuable samples of vegetatively propagated crops, which form an important part of many collections of genetic resources. Using the existing storage techniques for vegetatively propagated crops, they

can be most effectively preserved at ultra-low temperatures, employing cryopreservation in liquid nitrogen or its vapor; under such conditions, there is a complete cessation of metabolism in the plant tissues and cells. Cryopreservation requires minimal space and minimal maintenance [2] Cryopreservation methods have been developed for a large number of plant species [3]. In addition, the Commission on Genetic Resources for Food and Agriculture has released an updated Genebank Standard for Plant Genetic Resources for Food and Agriculture [4]. Cryopreservation methods are recognized as a biological tool for the long-term storage of plant genetic resources.

Currently, various methods of cryopreservation are being improved through techniques such as the vitrification method, encapsulation/dehydration method, and encapsulation/vitrification method [5]. Modified techniques have been developed, which further reduce the chance for lethal ice-crystal formation through the application of ultra-fast cooling and rewarming rates. These techniques are called the droplet vitrification method, V cryo-plate method, and D cryo-plate method. These methods are convenient to use for the cryopreservation of apexes (shoot tips) and meristems in vitro. Those cryopreserved by vitrification were: *Wasabia japonica* (wasabi), [6] and *Menta* L. (mint), [7]; by droplet vitrification: *Manihot esculenta* (cassava), [8] and *Musa* spp. [9]; by D cryo-plate: *Juncus effuses* (mat rush), [10].

Several cryopreservation techniques have been established on the basis of the conventional slow-freezing method. Initially, this method was used for the cryopreservation of apexes (shoot tips): *Rubus* spp. (raspberry, in vitro shoot tips, slow freezing) [11] and *Pyrus* spp. (pear, in vitro shoot tips, slow freezing) [12], but is now used for the cryopreservation of dormant buds of woody plants. This cryopreservation method is called the "Cryopreservation of dormant vegetative bud method". The method is now applied to the cryopreservation of red (*Ribes rubrum* L.) and black (*Ribes nigrum* L.) currants [13–15], other *Ribes* spp. (golden, clove, wax currant and gooseberries, [16]), and many woody plants: *Malus* spp. (apple), [17–23]; *Morus* spp. (mulberry), [24,25]; *Ulmus* spp. (elm), [26]; *Prunus cerasus* L., (cherry), [27]; *Fraxinus* spp., [28]; *Pyrus* L. (pear), [29]. Reliable estimates of the actual shelf life of material in liquid nitrogen are critical to efficient gene bank establishment. A high viability of dormant apple buds after 10 years of storage in liquid nitrogen vapor has been shown [30]. The percentage of live mulberry buds stored for 11.5 years in liquid nitrogen vapor was 98% [31]. Since the development of vitrification methods, several scientific publications have appeared indicating the exact viability and genetic stability of the materials after long-term cryostorage. Caswell and Kartha [32] demonstrated the possibility of in vitro cryopreservation of strawberry and pea meristems in LN for 28 years. In the case of strawberry meristems grown in vitro, no decrease in the percentage of viable meristems persisting for 8 weeks or 28 years was observed. This result is evidence that plant meristems can be stored in liquid nitrogen for long periods of time [32]. In addition, for wasabi shoots grown in vitro, there were no significant differences in the development and morphological characteristics between 10-year cryopreservation and 2-h cryopreservation. Wasabi plants obtained from the shoot tips cryopreserved for 10 years by vitrification were genetically stable [33].

Genetic stability was also confirmed using morphological parameters, flow cytometry measurements, and RFLP assays, suggesting that the cryopreservation method does not cause somaclonal variability as no significant differences were observed in regenerated material compared to the controls [33,34]. The metabolic stability of *Dioscorea deltoidea* and *Panax ginseng* after cryopreservation was also shown [35]. However, according to other authors, some genetic changes may occur in cryopreserved plants [36–38].

The cryostorage of peach collections has not been studied, but it is necessary to find a simpler and more reliable method to show the possibility of the cryostorage of collection samples by the method of dormant buds. In order to determine the suitability of this method, its capabilities and possible disadvantages, assess the influence of the variety and climatic conditions on the result of cryopreservation, for the first time, we undertook a detailed long-term study of all of these factors. Therefore, when carrying out our work

on placing accessions for long-term storage in liquid nitrogen vapor, the main method was the one for the cryopreservation of dormant vegetative buds, used for most fruit and berry crops. The main purpose of the study was to assess the influence of such factors as climatic conditions of the year of material sampling variety and storage conditions (low temperature and cryopreservation) on the viability of vegetative shoots of peach.

## 2. Materials and Methods

The material for the study were cuttings with dormant buds of five peach cultivars growing in the field gene bank at Krymsk Experiment and Breeding Station of VIR. These cultivars have different ripening period and winter hardiness (Table 1).

**Table 1.** Characteristics of the peach cultivars placed for storage at low- and ultra-low temperatures for 6 months (2019–2021).

| No. | Cultivar | VIR Catalogue No. | Winter Hardiness | Fruit Ripening Period |
|---|---|---|---|---|
| 1 | Baby Gold | k-40871 | medium | mid-late |
| 2 | Lucky 24 B | k-13305 | high | late |
| 3 | Lyubimets Krasnodara | k-40967 | medium | early |
| 4 | Podarok Kryma | k-41032 | medium | mid-early |
| 5 | Ustojchivyy 90 | k-43768 | high | late |

**'Baby Gold'** is a mid-late cultivar, bred in the USA. The fruit is mostly medium-sized (100–140 g), the pulp is orange, with a pleasant aroma, the stone does not separate. Winter hardiness is medium.

**'Lucky 24B'** the peach Fleming Fury (Flemin'Furi) PF Lucky 24B Yellow Peach—is a late cultivar of American breeding. Under the conditions of Krymsk, it ripens around August 25–30. The fruits are very large, elongated oval, with an average weight of 200 g, some up to 400 g or more. A red blush covers about 70% of the fruit surface. The pulp is fibrous, tender, yellow in color, the stone separates well. Very prolific. Winter hardiness is high.

**'Lyubimets Krasnodara'** is an early-ripening peach cultivar obtained from the free pollination of cv. 'Gayar-9′ in the North Caucasian Federal Scientific Center of Horticulture, Viticulture, Winemaking. The shape of the fruit is broad oval, the main color of the pulp is yellow, the skin is red. The pulp is fibrous. The stone does not separate. Winter hardiness is medium.

**'Podarok Kryma'** is a mid-early peach cultivar developed at the Nikita Botanical Gardens by I. N. Ryabov and A. N. Ryabova from crossing cvs 'Khidistavsky Bely' and 'Greensboro'. The trees are medium-sized, with a wide pyramidal crown, highly resistant to *Clasterosporium*, powdery mildew, and leaf curl. The fruits are round or broadly oval, the flesh is white, cartilaginous. They ripen in mid-August. The stone does not separate. Winter hardiness is medium.

**'Ustojchivyy 90'** is a late-ripening cultivar, resistant to curl and powdery mildew. The fruits are small (35–45 g), strongly pubescent, of mediocre taste. The pulp is cartilaginous, not juicy, white in color. Winter hardiness is high.

The cuttings were selected in the phase of winter plant dormancy in December 2019, 2020, and 2021 in the garden of Krymsk Experiment and Breeding Station of VIR.

Determination of the initial viability of peach cuttings and their cryopreservation was carried out such as the cryopreservation of black and red currant cuttings in previous works [14]. Namely, first, the cuttings were divided into segments 6–8 cm long, with 2–3 buds in a segment. The initial viability of the collected material was assessed by growing 10 cuttings, with three replications per cultivar in the glass containers with water, under $21 \pm 1$ °C, 16 h light/8 h dark, until the formation of the leaves and roots. A part of the cutting was left as a reference and stored in a HUURRE refrigerator at $-5$ °C, while the larger part of the plant material was dried at $-4$ °C down to the required moisture

in the plants, 28–32%. After drying, the cuttings were gradually frozen in foil laminated packages using a multistep technique. Freezing to −30 °C was carried out at a rate of 1–2 °C per min. At −30 °C, the cuttings were kept for 30 min. Then, the cuttings were frozen to a temperature from −48 to −50 °C at a rate of 3–4 °C per min. The frozen samples were placed into cryopreservation tanks for long-term storage in liquid nitrogen vapor at a temperature from −183 to −185 °C for six months. In the spring, the cuttings were removed from the tanks, defrosted in a water bath, and their viability was determined. At the same time, cuttings were analyzed, which were stored in the refrigerator at –5 °C. The viability of both frozen and refrigerated cuttings was assessed by growing 10 cuttings with three replicates for each cultivar.

*Climate Conditions in 2019–2021*

Krymsk is located at Krasnodar Krai, Russia (coordinates: 44°55′24″ N, 37°58′50″ E) and has a humid subtropical climate.

The weather conditions of the experiment during the years of the study were characterized by increased sums of active temperatures above 10 °C (3850, 3890, and 3560 °C in 2019, 2020, and 2021, respectively) compared with the long-term norm of 1971–2000 (3460 °C) (Figure 1a). The minimum temperature of the month when the cuttings were collected (December) was observed in 2021 (–11.7 °C); in 2019. the minimum temperature of December was –5.4 °C; and in 2020, it was –6.7 °C (Figure 1b). The largest amount of precipitation for the period with temperatures above 10 °C was in 2021 (563 mm); the smallest in 2020 (210 mm); and in 2019 (320 mm), it was close to the long-term norm (327 mm) (Figure 1c).

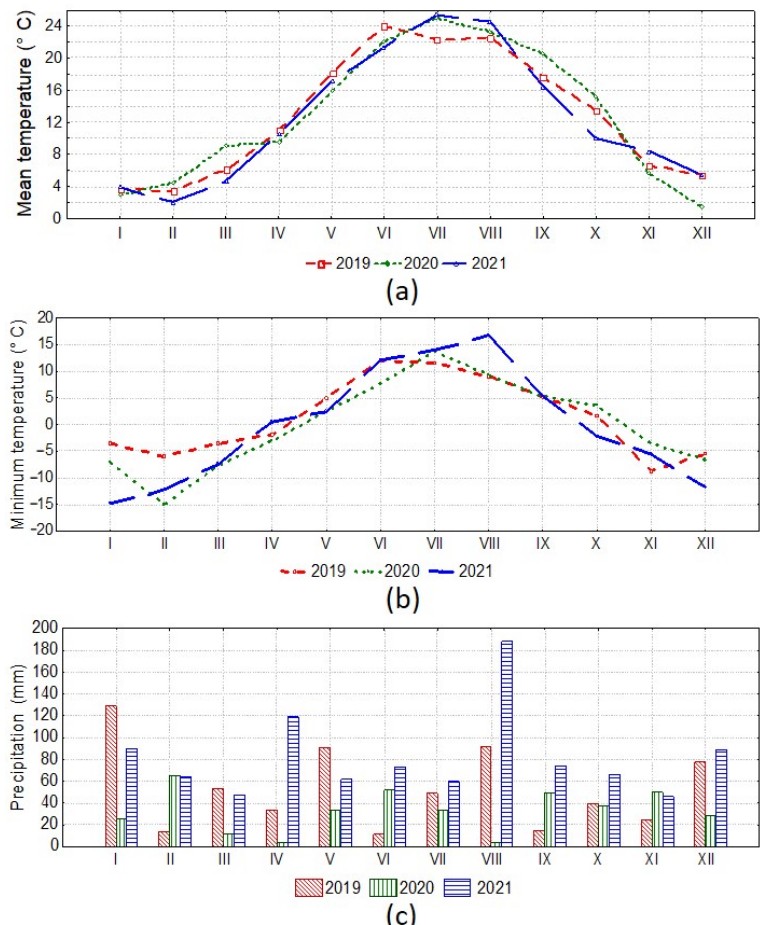

**Figure 1.** Weather conditions of the experiment. (**a**) Average monthly temperature; (**b**) the minimum temperature of the month; (**c**) monthly total precipitation.

In a full factorial experiment, the influence of three factors on the viability was studied: the method of storage, the year of collecting cuttings, and the variety. Each experiment was performed in triplicate. The effect of the factors was studied by the analysis of variance in the Statistica 13.3 package. A posteriori analysis was carried out according to Tukey's test. The study adopted a significance level of 5%.

### 3. Results and Discussion

The three-factor analysis of variance showed that the viability of peach cuttings was significantly affected by all three studied factors (Figure 2): storage method ($p < 0.001$), variety ($p < 0.001$), and year ($p < 0.001$). The interaction of factors was insignificant. The main contribution to the change in viability was made by the method of storage (86.2%). The influence of the variety was many times smaller (3.1%), and that of the year was even less (1.9%).

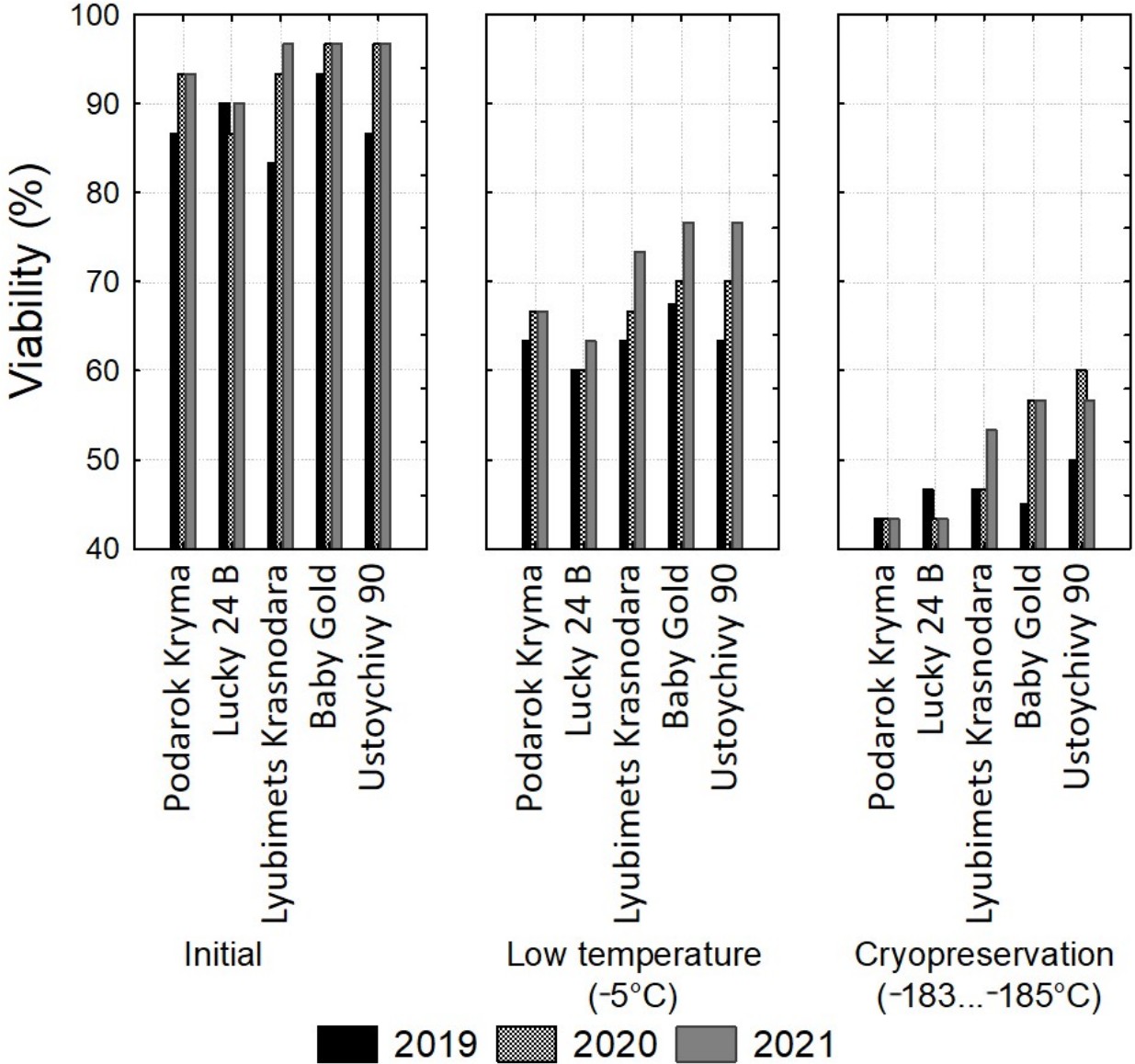

**Figure 2.** Viability of the peach cultivars preserved by different storage methods (2019–2021). The varieties are arranged in ascending order by average viability after cryopreservation.

**Effect of the storage method**. For three years, the initial viability averaged 92.0% for the cultivars. After low-temperature storage, it significantly decreased by an average of

24.9% and amounted to 67.1%. After cryopreservation, it decreased by an average of 17.8% and amounted to 49.3%. The same decrease in viability depending on the storage method was observed in [14,22,39].

**Effect of the year of the experiment**. In general, the viability in 2020 for all types of storage was higher than in 2019, and in 2021, it was higher than in 2020 (Figure 3). The effect of the year was considered separately for different types of storage due to the significant difference between them. The year had a significant effect on the initial viability ($p = 0.032$) and viability after low-temperature storage ($p = 0.043$), but its effect on the results of cryopreservation was not significant ($p = 0.676$) (Figure 3). Different initial viability levels after low-temperature storage may be explained by different degrees of plant hardening at the time of collecting cuttings due to a higher minimum temperature in December in 2019 compared to 2020, and an even lower one in 2021; at the same time, pretreatment before cryopreservation (dehydration at –5 °C, and slow freezing before placement into liquid nitrogen) reduced the effect of the year on the efficiency of cryopreservation. In the works by Pathirana, R. et al. (2018) and Vogiatzi C. et al. (2011) and Jenderek M.M. et al. (2020), similar results were obtained for apple-tree cuttings and *Salix* dormant buds [40–42].

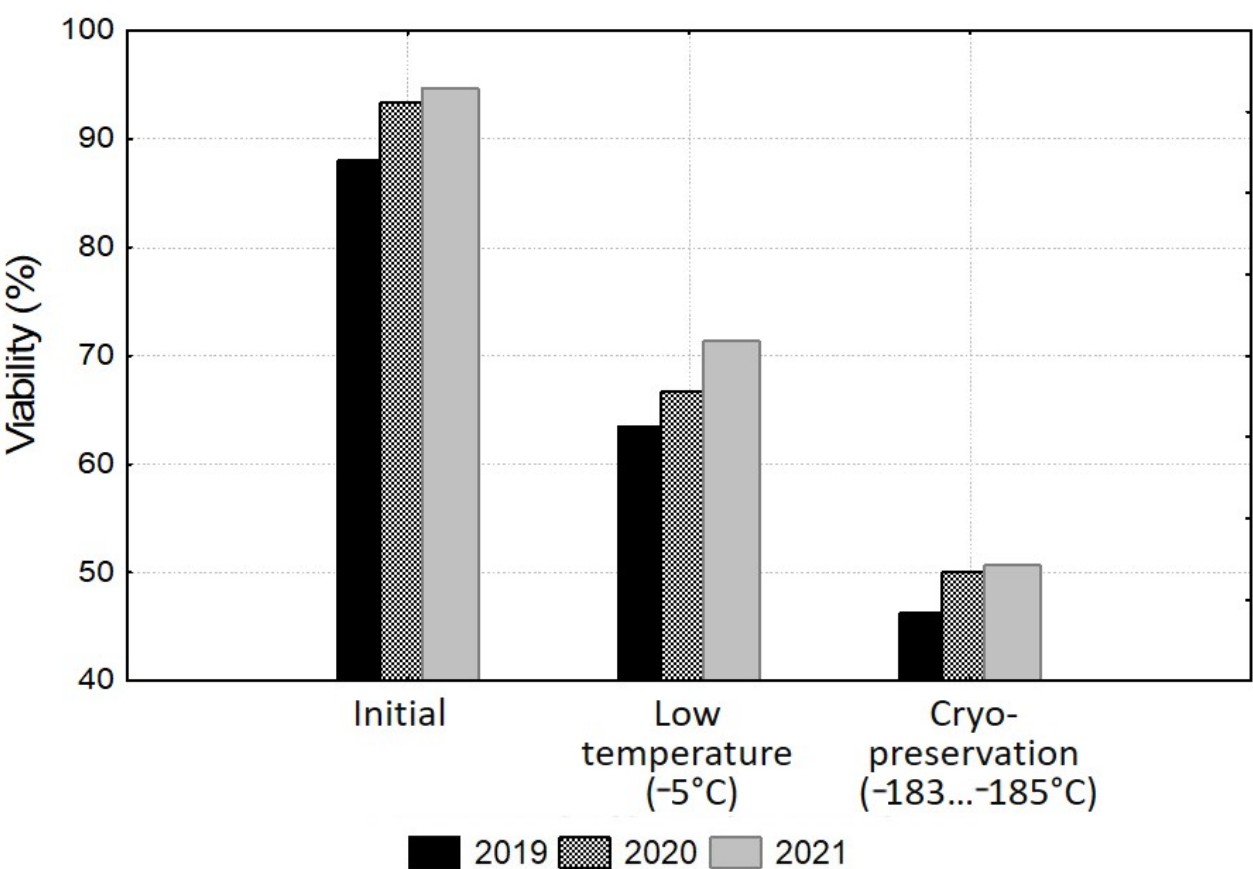

**Figure 3.** The effect of the experimental year and storage method on the average viability of the peach cultivars.

**The variety features** were analyzed separately for different years and storage methods. In 2019, the initial viability averaged 88.0% for the cultivars (Table 2). The after low-temperature storage amounted to 63.3%, and the after cryopreservation amounted to 47.3%. There were no significant differences between the varieties in each variant for any storage method ($p \geq 0.485$).

**Table 2.** The effect of low-temperature storage (−5 °C) and cryopreservation in liquid nitrogen vapor (−183 . . . −185 °C) on the viability of peach cuttings when assessed in laboratory conditions (2019) *.

| No. | Cultivar | VIR Catalogue No. | Viability of Peach Cuttings with Dormant Vegetative Buds, % | | |
|---|---|---|---|---|---|
| | | | Initial | After Storage under at −5 °C | After Cryopreservation under −183 °C . . . −185 °C |
| 1 | Podarok Kryma | k-41032 | 86.7 ± 3.3 [ijkl] | 63.3 ± 3.3 [bcdefg] | 43.3 ± 3.3 [a] |
| 2 | Lucky 24 B | k-13305 | 90.0 ± 5.8 [jkl] | 60.0 ± 0.0 [abcdefg] | 46.7 ± 3.3 [ab] |
| 3 | Lyubimets Krasnodara | k-40967 | 83.3 ± 3.3 [hijkl] | 63.3 ± 3.3 [bcdefg] | 46.7 ± 3.3 [ab] |
| 4 | Baby Gold | k-40871 | 93.3 ± 3.3 [kl] | 66.7 ± 3.3 [defgh] | 50.0 ± 5.8 [abc] |
| 5 | Ustojchivyy 90 | k-43768 | 86.7 ± 3.3 [ijkl] | 63.3 ± 3.3 [bcdefg] | 50.0 ± 5.8 [abcd] |
| | Mean value | | 88.0 ± 1.7 | 63.3 ± 1.1 | 47.3 ± 1.2 |

* The same letters mark the average values that do not differ significantly for $p < 0.05$.

In 2020, the initial viability averaged 93.3% for the cultivars (Table 3), the after low-temperature storage amounted to 66.7%, while the after cryostorage amounted to 50.0%. The cultivars did not differ in the initial viability and in viability after low-temperature storage, but differed in the percentage of viability after cryopreservation ($p = 0.023$). Cvs. 'Podarok Kryma' and 'Lucky 24 B' had a significantly lower viability percentage (43.3%) after cryopreservation than cv. 'Ustojchivy 90′ (60.0%). The remaining cultivars had intermediate values of viability and did not differ significantly from the contrasting cultivars.

**Table 3.** The effect of low-temperature storage (−5 °C) and cryopreservation in liquid nitrogen vapor (−183 . . . −185 °C) on the viability of peach cuttings when assessed in laboratory conditions (2020) *.

| No. | Cultivar | VIR Catalogue No. | Viability of Peach Cuttings with Dormant Vegetative Buds, % | | |
|---|---|---|---|---|---|
| | | | Initial | After Storage under at −5 °C | After Cryopreservation under −183 °C . . . −185 °C |
| 1 | Podarok Kryma | k-41032 | 93.3 ± 3.3 [kl] | 66.7 ± 3.3 [cdefgh] | 43.3 ± 3.3 [a] |
| 2 | Lucky 24 B | k-13305 | 86.7 ± 3.3 [ijkl] | 60.0 ± 0.0 [abcdefg] | 43.3 ± 3.3 [a] |
| 3 | Lyubimets Krasnodara | k-40967 | 93.3 ± 3.3 [kl] | 66.7 ± 3.3 [cdefgh] | 46.7 ± 3.3 [ab] |
| 4 | Baby Gold | k-40871 | 96.7 ± 3.3 [l] | 70.0 ± 0.0 [efghi] | 56.7 ± 3.3 [abcdef] |
| 5 | Ustojchivyy 90 | k-43768 | 96.7 ± 3.3 [l] | 70.0 ± 0.0 [efghi] | 60.0 ± 0.0 [bcdefg] |
| | Mean value | | 93.3 ± 1.8 | 66.7 ± 1.8 | 50.0 ± 3.5 |

* The same letters mark the average values that do not differ significantly for $p < 0.05$.

In 2021, the initial viability averaged 94.7% for the cultivars (Table 4). After low-temperature storage, it amounted to 71.3%, while after cryopreservation, it amounted to 50.7%. The analysis of variance showed a significant difference in the viability of the cultivars only after cryopreservation ($p = 0.030$), and there were no differences in the initial viability and the viability after low-temperature storage. Tukey's test did not confirm differences between cultivars after cryopreservation.

**Table 4.** The effect of low-temperature storage (−5 °C) and cryopreservation in liquid nitrogen vapor (−183 . . . −185 °C) on the viability of peach cuttings when assessed in laboratory conditions (2021) *.

| No. | Cultivar | VIR Catalogue No. | Viability of Peach Cuttings with Dormant Vegetative Buds, % | | |
| --- | --- | --- | --- | --- | --- |
| | | | Initial | After Storage under −5 °C | After Cryopreservation under −183 °C . . . −185 °C |
| 1 | Podarok Kryma | k-41032 | 93.3 ± 3.3 [kl] | 66.7 ± 3.3 [cdefgh] | 43.3 ± 3.3 [a] |
| 2 | Lucky 24 B | k-13305 | 90 ± 5.8 [jkl] | 63.3 ± 3.3 [bcdefg] | 43.3 ± 3.3 [a] |
| 3 | Lyubimets Krasnodara | k-40967 | 96.7 ± 3.3 [l] | 73.3 ± 3.3 [fghij] | 53.3 ± 3.3 [abcde] |
| 4 | Baby Gold | k-40871 | 96.7 ± 3.3 [l] | 76.7 ± 3.3 [ghijk] | 56.7 ± 3.3 a[bcdef] |
| 5 | Ustojchivyy 90 | k-43768 | 96.7 ± 3.3 [l] | 76.7 ± 3.3 [ghijk] | 56.7 ± 3.3 [abcdef] |
| | Mean value | | 94.7 ± 1.3 | 71.3 ± 2.7 | 50.7 ± 3.1 |

* The same letters mark the average values that do not differ significantly for $p < 0.05$.

According to the three-year average characteristics of the cultivars, the analysis of variance showed a significant difference in the viability of the cultivars (Table 5) only after cryopreservation ($p = 0.004$). There were no differences among the cultivars in the initial viability and the viability after low-temperature storage. According to the results of the three years of study, cvs. 'Podarok Kryma' (43.3%) and 'Lucky 24 B' (44.4%) showed the lowest viability after cryopreservation, significantly lower than cvs. 'Baby Gold' (54.4%) and 'Ustojchivy 90′ (55.6%). Cv. 'Lyubimets Krasnodara' (48.9%) occupied an intermediate position. A tendency was observed toward higher viability after cryopreservation in the cultivars with higher initial viability: the correlation coefficient between these indicators was 0.84, but it was not significant due to a small sample; viability after low-temperature storage significantly correlated with the initial viability (0.93) (i.e., the result of cryopreservation and low-temperature storage of peach cuttings can be largely determined by the viability of the initial material). The response of the genotype to the impact of ultra-low temperatures was also traced in the work by Verzhuk V. et al. (2018) [43]. Such a response can be either neutral or negative. A positive response was observed in other materials such as pollen from fruit crops Verzhuk V.G. et al. (2005); Pavlov A.V. et al. (2019) [13,44].

**Table 5.** The effect of low-temperature storage and cryopreservation in liquid nitrogen vapor on the viability of peach cuttings when assessed under laboratory conditions (2019–2021) (summary, average value over 3 years).

| No. | Cultivar | VIR Catalogue No. | Viability of Peach Cuttings with Dormant Vegetative Buds, % | | |
| --- | --- | --- | --- | --- | --- |
| | | | Initial | After Storage under at −5 °C | After Cryopreservation under −183 °C . . . −185 °C |
| 1 | Podarok Kryma | k-41032 | 91.1 ± 2.2 [g] | 65.6 ± 1.1 [ef] | 43.3 ± 0.0 [a] |
| 2 | Lucky 24 B | k-13305 | 88.9 ± 1.1 [g] | 61.1 ± 1.1 [de] | 44.4 ± 1.1 [ab] |
| 3 | Lyubimets Krasnodara | k-40967 | 91.1 ± 4.0 [g] | 67.8 ± 2.9 [ef] | 48.9 ± 2.2 [abc] |
| 4 | Baby Gold | k-40871 | 95.6 ± 1.1 [g] | 71.1 ± 2.9 [ef] | 54.4 ± 2.2 [bcd] |
| 5 | Ustojchivyy 90 | k-43768 | 93.3 ± 3.3 [g] | 70.0 ± 3.8 [ef] | 55.6 ± 2.9 [cd] |
| | Mean value | | 92.0 ± 1.1 | 67.1 ± 1.8 | 49.3 ± 2.5 |

The same letters mark average values that do not differ significantly for $p < 0.05$.

It should be noted that for long-term storage of the peach gene pool, the method of cryopreservation of dormant buds is not inferior in efficiency to the methods of encapsulation–dehydration [45] and vitrification of the peach shoot tips [46]. In the first case, the viability of the peach shoot tips after cryopreservation was 33–36%; in the second case, it was 60%, which is comparable with our results—43.3–55.6% of viable buds after cryopreservation. At the same time, the method of stepwise freezing of dormant buds requires much less labor and reagents than the mentioned methods. After cryopreservation by the method of dormant buds of other fruit crops such as apple [19,20] and red and black currants [13,14], higher values of viability percentages were obtained: in apple 84–90%, in red and black currants 58.9–73.5% compared with the viability of peach buds after cryopreservation. This could perhaps be because peach is a more thermophilic crop; although more winter-hardy varieties were selected from the collection for study, the peach is a more difficult material for cryopreservation.

## 4. Conclusions

The three-factor analysis of variance showed that the viability of the peach cuttings was significantly affected by all three studied factors: storage method, variety, and year.

It was shown that the method of the cryopreservation of dormant vegetative buds is simple and effective and is well-suited for long-term storage of the peach gene pool. The viability of the peach buds in all of the studied cultivars was in the range from $43.3 \pm 0.0\%$ to $55.6 \pm 2.9\%$, which exceeded the minimum requirement for samples subjected to long-term cryogenic storage in a cryobank.

The applied cryopreservation protocol was effective for the peach buds.

After low-temperature storage, the viability of the peach buds was slightly higher than after cryostorage; the result depended on the initial state of the material.

Low-temperature storage of peach cuttings at –5 °C can be used for short-term preservation.

The year of material collection had a significant effect on the initial viability ($p = 0.032$) and viability after low-temperature storage ($p = 0.043$); the effect on viability after cryopreservation storage was insignificant ($p = 0.676$).

According to the three-year average characteristics of the cultivars, the analysis of variance showed a significant difference in the viability of the cultivars only after cryogenic storage ($p = 0.004$). There were no differences in the initial viability and the viability after low-temperature storage ($p = 0.485$ and $p = 0.132$, respectively).

**Author Contributions:** Conceptualization: V.V.; Experiment performance: V.V. and A.P.; Data analysis and visualization: L.Y.N.; Resources, writing—original draft preparation: V.E., T.G., O.E. and M.S.; Writing—review and editing: G.F. All authors have read and agreed to the published version of the manuscript.

**Funding:** This research received no external funding.

**Institutional Review Board Statement:** Not applicable.

**Data Availability Statement:** The data presented in this study are available on request from the corresponding author.

**Acknowledgments:** The research was performed within the framework of the State Task according to the theme plan of VIR, Project No. 0481-2022-0004 "Improving the approaches and methods for ex situ conservation of the identified genetic diversity of vegetatively propagated crops and their wild relatives, and development of technologies for their effective utilization in plant breeding".

**Conflicts of Interest:** The authors declare no conflict of interest.

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
