# Peer review of "Post-Cryogenic Viability of Peach (Persica vulgaris Mill.) Dormant Buds from the VIR Genetic Collection"

_agriculture, doi:10.3390/agriculture13010111_

Round 1

Reviewer 1 Report

the work here is meaningful. but ”only 10 cuttings were used in every replication”,and” A part of the cuttings was left as reference and stored in a HUURRE refrigerator at 5 °C”(line 139-143). So the sample in every treatment was not enough and inaccurated. And there was no Multiple comparison among the levels.  So the results were not scientific and statistical enough.

Author Response

Dear Reviewer!

Thank you for carefully reading our article. We have made changes in the article according to your comments.

Comments and Suggestions for Authors

The work here is meaningful. but ”only 10 cuttings were used in every replication”,and” A part of the cuttings was left as reference and stored in a HUURRE refrigerator at −5 °C”(line 139-143). So the sample in every treatment was not enough and inaccurated. And there was no Multiple comparison among the levels.  So the results were not scientific and statistical enough.

Answer: The viability of the collected material was assessed by growing 10 cuttings, with three replications (Lines 144-145, 156-158). Three-way ANOVA with Tukey HSD was used for multiple compsrison (Lines 177-181). The markers for homogeneous groups was added to all tables.

Reviewer 2 Report

- Statistical grouping letters should be given in all tables.

- The title of Table 2 is not written.

- In Table 5, the title of the 1st column is not read.

- The conclusion is long. It should be written shorter and clearer.

- The statistics information in the conclusion section should be removed. The focus should be directly on the result.

Author Response

Dear Reviewer!

Thank you for carefully reading our article. We have made changes in the article according to your comments.

Questions: Statistical grouping letters should be given in all tables; The title of Table 2 is not written; In Table 5, the title of the 1st column is not read.

Answers: The markers for homogeneous groups was added to all tables, the title of Table 2 is written, In Table 5, the title of the 1st column was corrected.

Question: The conclusion is long. It should be written shorter and clearer.

Answer:  The conclusion was shortened and edited. ( lines 289-297)

Question: The statistics information in the conclusion section should be removed. The focus should be directly on the result.

Answer:  The statistics information in the conclusion section was removed.

Reviewer 3 Report

1. The introduction is too long and does not fit the topic of the paper, and does not condense the scientific question well. Please use concise language to summarize the research background of the article, the current research status at home and abroad, and the problem to be solved in this paper.

2. The innovation of the manuscript is average, and it does not reflect the unique innovation with other research scholars. It is recommended to add an innovative description of the manuscript.

3. In Results and Discussion, the authors have cited a large number of references to prove the same results. However, this paper lack of own in-depth data analysis, and fails to reveal the reason of study results.

4. Please summarize the lengthy conclusion concisely.

Author Response

Dear Reviewer!

Thank you for carefully reading our article. We have made changes in the article according to your comments.

 Question 1: The introduction is too long and does not fit the topic of the paper, and does not condense the scientific question well. Please use concise language to summarize the research background of the article, the current research status at home and abroad, and the problem to be solved in this paper.

Answer 1: We have shortened and edited the introduction. (lines 101 – 111)

Question 2: The innovation of the manuscript is average, and it does not reflect the unique innovation with other research scholars. It is recommended to add an innovative description of the manuscript

Answer 2:  unique innovation - for the first time the method of cryopreservation of dormant buds for peach was applied and for the first time the influence of variety and climate and storage method on the post-cryogenic viability of peach was studied (101-106)

Question 3: In Results and Discussion, the authors have cited a large number of references to prove the same results. However, this paper lack of own in-depth data analysis, and fails to reveal the reason of study results.

Answer 3: We have added to the discussion a comparative analysis of the viability of other fruits after cryo, because. We believe that the main novelty of our study is the demonstration of the possibility of storing such a heat-loving crop as peach in cryo with the achievement of minimum performance in accordance with storage standards (289 -297)

Question 4: Please summarize the lengthy conclusion concisely.

Answer 4: We was summarize the lengthy conclusion concisely. (lines 284 - 296)

Round 2

Reviewer 1 Report

Accept after minor revision (corrections to minor methodological errors and text editing)

Reviewer 3 Report

This paper is revised well.